# The association between water intake and future cardiometabolic disease outcomes in the Malmö Diet and Cancer cardiovascular cohort

Harriet A. Carroll[1,2]*, Ulrika Ericson[1], Filip Ottosson[1], Sofia Enhörning[1], Olle Melander[1]

1 Clinical Research Centre, Cardiovascular Research—Hypertension, Lund University, Malmö, Sweden,
2 School of Medicine, Medical Sciences and Nutrition, University of Aberdeen, Aberdeen, United Kingdom

* harriet.carroll@med.lu.se

## Abstract

The aim of this study was to explore the longitudinal association between reported baseline water intake and incidence of coronary artery disease (CAD) and type 2 diabetes in the Malmö Diet and Cancer Cohort (n = 25,369). Using cox proportional hazards models, we separately modelled the effect of plain and total (all water, including from food) water on CAD and type 2 diabetes risk, whilst adjusting for age, sex, diet collection method, season, smoking status, alcohol intake, physical activity, education level, energy intake, energy mis-reporting, body mass index, hypertension, lipid lowering medication, apolipoprotein A, apolipoprotein B, and dietary variables. Sensitivity analyses were run to assess validity. After adjustment, no association was found between tertiles of plain or total water intake and type 2 diabetes risk. For CAD, no association was found comparing moderate to low intake tertiles from plain or total water, however, risk of CAD increased by 12% (95% CI 1.03, 1.21) when comparing high to low intake tertiles of plain water, and by 17% (95% CI 1.07, 1.27) for high *versus* low tertiles of total water. Sensitivity analyses were largely in agreement. Overall, baseline water intake was not associated with future type 2 diabetes risk, whilst CAD risk was higher with higher water intakes. Our findings are discordant with prevailing literature suggesting higher water intakes should reduce cardiometabolic risk. These findings may be an artefact of limitations within the study, but future research is needed to understand if there is a causal underpinning.

## Introduction

Hydration status and fluid intake are often cited as important factors in influencing several health outcomes, such as gluco-regulatory and cardiovascular health [1, 2]. There is value in understanding this relationship considering the accessibility of increasing fluid intake compared to other dietary interventions.

In terms of gluco-regulatory health, previous findings have been mixed, with possible hints towards sex-differences. Roussel *et al.* [3] found a non-linear relationship between

**Data Availability Statement:** Due to ethical and legal restrictions related to the Swedish Biobanks in Medical Care Act (2002:297) and the Personal Data Act (1998:204) regarding deposition of data,

data are available upon request via the authors or by contacting registrator@lu.se. Lund University, represented by the authors, is the authority obliged to follow Swedish legislation and can be contacted at registrator@lu.se.

**Funding:** This project was funded by the Esther Olssons stiftelse II & Anna Jönssons Minnesfond (HC). The funders had no role in study design, data collection and analysis, decision to publish, or preparation of the manuscript.

**Competing interests:** HAC has received research funding from the Economic and Social Research Council, the European Hydration Institute, and the Esther Olssons stiftelse II & Anna Jönssons Minnesfond; has conducted research for Tate & Lyle; and has received speakers fees from Danone Nutricia Research. HAC, SE and OM have received conference, travel and accommodation fees from Danone Nutricia Research. O.M. has received a research grant for another study and consultancy fees from Danone Nutricia Research. No other authors declare any conflicts of interest. This does not alter our adherence to PLOS ONE policies on sharing data and materials.

higher water intake and lower risk of hyperglycaemia in a French cohort. Carroll *et al.* [4] supported these findings in a small UK cohort, demonstrating lower diabetes risk score being associated with increased water intake, whilst in a Swedish cohort, high urine osmolality and copeptin (surrogate marker of arginine vasopressin [AVP]) were associated with higher fasting plasma glucose concentrations [5]. However, in an exclusively female US cohort, no association was found between water intake and type 2 diabetes risk [6]. Such findings may be explained by sex differences; indeed in a UK cohort, higher plain water intake was associated with lower risk of elevated glycated haemoglobin in men but not women [4].

In terms of cardiovascular health, plasma osmolality has been shown to be higher (which may indicate lower fluid intake) in patients presenting with stroke *versus* healthy controls [7]. In the Adventist Health Study, high and moderate water intake was associated with lower risk of fatal coronary heart disease in men and women, respectively [8]. Conversely, Sontrop *et al.* [9] found no association between water intake and cardiovascular disease, and Leurs *et al.* [10] found no association between total fluid or water intake and ischemic heart disease or stroke mortality in neither men nor women.

Many mechanisms have been proposed to explain the link between low water intake and increased risk of cardiometabolic disease, such as cell volume-mediated hormone signalling, blood viscosity, and decreased satiety resulting in positive energy balance [8, 11, 12]. The prevailing mechanism much research has focused on is elevations in AVP induced from low fluid intake. Elevated AVP theoretically could increase hepatic glucose output via acting on the hypothalamic-pituitary-adrenal axis resulting in higher cortisol levels via V1b receptors [13]; we have previously proposed that high AVP caused from low fluid intake could elicit a Cushing's syndrome-like phenotype due to the lack of negative feedback loop in AVP-mediated adrenocorticotropic hormone release [14]. Accordingly, consuming water to reduce AVP should help to inhibit this circuit. Further, AVP acts on V1a receptors which may mediate glycogenolysis, gluconeogenesis, and glucagon release, thus resulting in insulin resistance [15], all implicated in metabolic and cardiovascular health.

Indeed, consuming additional water has been shown to reduce circulating AVP (measured by its surrogate marker, copeptin) [16, 17], and thus may have therapeutic potential. Further, several studies have repeatedly shown predictive value of copeptin (a stable biomarker of AVP) in relation to the metabolic syndrome, type 2 diabetes, and cardiovascular diseases [5, 18–21]. However, methodologies with stronger causal inference have provided less certainty, with some showing improved glycaemia with increasing water to reduce AVP/copeptin [22], whilst others showing no effect [16, 23]. Such controlled studies have thus far been conducted in short-term settings only, potentially explaining the lack of obvious health benefits which may occur only after long-term exposure.

Therefore, there is much to understand regarding the relationship between water intake and cardiometabolic health. Accordingly, the aim of this study is to explore the association between reported water intake at a baseline exam and incidence of coronary artery disease (CAD) and type 2 diabetes during long-term follow-up in the Malmö Diet and Cancer Cohort (MDC). We hypothesise that higher water intakes will be associated with lower CAD and type 2 diabetes risk.

## Methods

### Study design and population

The present analyses use data from the MDC which is a prospective cohort from Sweden. All men born between 1923 and 1945 and all women born between 1923 and 1950, fluent in

Swedish, and residing in Malmö from 1991–1995 were eligible to participate. Participants were chosen via random selection using the municipal registry along with local advertisements and recommendations from other participants. Complete baseline measures were taken between January 1991 and September 1996 from n = 28,098 study volunteers. Detailed descriptions of the cohort have previously been published [24, 25]. At baseline anthropometrics were measured; approximately two weeks later, questionnaires were completed at home and an interview was conducted to complete dietary data collection. A baseline blood draw was taken from a subset of ~6000 participants which identified prevalent type 2 diabetes. The study was approved by the Ethical committee at Lund University (LU-51-90) and written consent was obtained by study participants.

## Measure of diet

Dietary data were collected via a modified diet history method designed to capture both current and usual diet (described previously [26]), which included: (i) a 7-day food diary mainly covering lunch and dinner meals; (ii) a 168-item semi-quantitative diet history food frequency questionnaire (FFQ) of regularly eaten foods outside the main meals and not covered by the food diary; (iii) a 45–60 min diet history interview including questions about usual portion sizes, food preparation practices, and to check for overlap between the food diary and the FFQ. The validity of these methods has previously been reported [27]. These data were summarised and converted to nutrient intakes based upon the Swedish Food Database PC KOST2-93. The coding routines were slightly changed in Sept 1994 to shorten the interview time. The change did not have any major influence on the ranking on individuals on dietary intakes [26]. For the present analyses, water was defined in two different ways, with intakes split into tertiles to determine low, moderate, and high intakes for analyses:

1. Plain water: sum of mineral and tap water intake (average data intake recorded in the food diary)

2. Total water: reported average daily intake of all foods and beverages ([intake frequencies recorded in the 7-day food diary × portion sizes from the diary or diet history interview] + [intake frequencies × portion sizes from the FFQ]) was converted to water intake by multiplying with factors for water content obtained from PC KOST2-93 and summed

For model adjustment, non-water beverages was also used which included all beverages except plain water (as defined above).

A variable to identify likely energy misreporting was created, based upon having an energy intake to estimated basal metabolic rate ratio outside the 95% confident interval (CI) limits of the estimated physical activity level (from a structured questionnaire), using logistic regression to assess the risk of being a low- or high-energy reporter [28].

## Measure of disease

We used record linkage with national registries to retrieve prevalent and incident CAD events (i.e. fatal and non-fatal myocardial infarction, death due to coronary heart disease, or revascularization procedures) as described previously [29]. Diabetes was determined either via questionnaire data on a physician diagnosis of diabetes, being on antidiabetic medication, having a fasting blood glucose $\geq 6.1$ mmol·$L^{-1}$ (which corresponds to a fasting plasma glucose of 7.0 mmol, $L^{-1}$) in the subsample who had a baseline blood sample, or through record linkage with national or local diabetes registries, as described previously [30].

### Measure of other variables

Demographic and characteristic data, such as smoking and age, were collected by validated questionnaires at baseline. Current smoking was defined as any smoking within the last year. Blood pressure was measured twice at the baseline visit using a mercury column sphygmomanometer after 5 min rest in the cumbent position. Physical activity was measured using a modified Minnesota leisure time physical activity questionnaire that was later validated within the MDC cohort [31]. For the present analysis, physical activity levels were divided into quintiles of activity. Height, weight, waist, and hip circumference were measured without shoes and rounded to the nearest cm or 0.1 kg (as appropriate). In a subsample, 45 mL blood was drawn in an overnight fasted state, and plasma and serum supernatants were frozen at -80 °C after centrifugation [24].

### Statistical analysis

Participants who had a history of CAD or type 2 diabetes (meeting any diabetes criteria, as described above) at baseline (in order to assess the relationship with new-onset disease risk), or had missing data of exposures relevant to the analyses, were excluded from further analyses (Fig 1). Based on previous work suggesting potential sex differences [11], an interaction effect was tested by adding a multiplicative factor to the model (sex*tertile water intake treated as continuous).

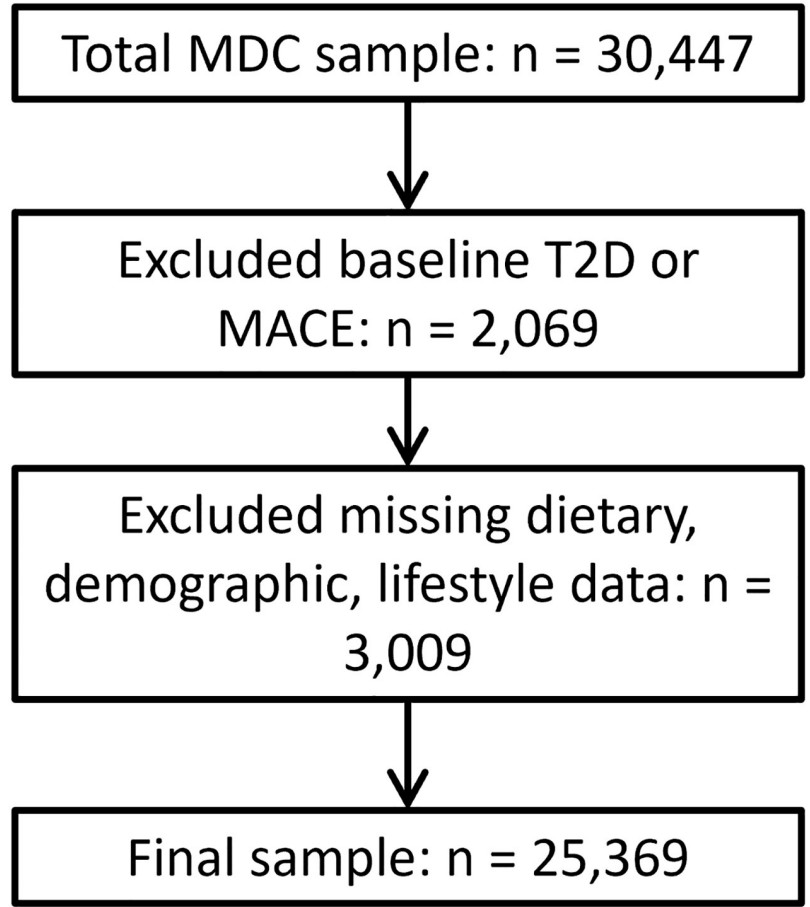

**Fig 1. Flow chart of sample size.**

Descriptive statistics of participant characteristics and dietary intake are presented as the entire sample, and according to tertiles (median [interquartile range, IQR] plain water intakes: low: 199 [93, 298]; moderate: 564 [474, 662]; high: 1140 [945, 1506] mL/d) of plain water. Data were non-normally distributed (according to PP and QQ plots, and Kolmogorov-Smirnov test); as such data are presented as medians and IQR for continuous variables, or percentage of the sample for categorical variables. Kruskal-Wallis (with Dunn-Bonferroni *post hoc* comparisons) or Chi square were used to analyse descriptive data. Linear trends for continuous descriptive data were assessed with a univariate general linear model. Pearson's *r* was used to investigate correlations between different water intakes (plain and total water) and dietary factors.

Cox proportional hazards model was used to estimate the hazard ratio (HR) of tertiles of plain and total water intake to CAD or type 2 diabetes incidence. Tertiles were used to avoid outliers at the upper intakes skewing the coefficients. Models were run without adjustment (model 1), followed by additional adjustment for: age, sex, diet method version, season (model 2); smoking, alcohol intake, physical activity level, education (model 3); energy intake, energy misreporting category, BMI, hypertension, lipid lowering medication, apolipoprotein A, apolipoprotein B (model 4); processed meat, wholegrains, and for plain water only: total water minus plain water (to account for the hydrating properties of non-plain water fluid). All variables were checked for the proportional hazards assumption by including a time-dependent covariate. Some variables violated this assumption. To test their impact, we re-ran the analyses omitting these variables, and also stratified the analyses by these variables (pre-defined categories or median split). Data were analysed using SPSS (version 27, IBM), with $\alpha = 0.05$.

Sensitivity analyses additionally analysed:

(i) plain and total water in cups (240 mL) per day (as per Carroll *et al.*, 2016 [11]) (as categorisation can reduce power); (ii) model without reporter category adjustment (in case there was collinearity with other included variables such as BMI and as reporting accuracy could not be considered as a confounder as it does not influence disease development); (iii) plausible energy reporters only (to reduce the chance that those more likely reporting inaccurately were influencing results); (iv) removal of the top 10% of fluid intakes (to account for potential illness-induced polydipsia and misreporters); (v) removal of those with type 2 diabetes/CAD within 2 years of baseline measurements (in case recent awareness of ill-health/diagnosis altered health behaviours); (vi) inclusion of hypertensive participants only (to see whether pathology could be influencing results); (vii) inclusion of normotensive participants only (to see whether pathology could be influencing results); (viii) energy adjusted dietary variables (to ensure results were not skewed by relative intake levels); (ix) excluding diet changers (to remove the influence of known dietary changes which could add noise to the data). Sensitivity analyses are presented in the S1 Table.

## Results

Mean (± standard deviation) follow-up time to incident diagnosis was 20 ± 7 (median 23, IQR 17, 25) years for CAD, and 20 ± 7 (median 23, IQR 15, 25) years for type 2 diabetes. A total of n = 25,369 participants were included in these analyses (n = 15,789 women). Out of these, 14.7% had new onset CAD, and 16.7% had new onset type 2 diabetes during follow-up. Tertiles of plain water intake represent low, moderate, and high intakes and are defined as a median intake of 199 (IQR 93, 298), 564 (IQR 474, 662), and 1140 (IQR 945, 1506) g/d, respectively. There was no sex interaction, so analyses were not sex-stratified. Table 1 shows baseline participant data overall and split by tertiles of plain water intake. The average (median) age was 57 (IQR 51, 64) years, and median BMI was 25.1 (IQR 22.9, 27.7) kg/m$^2$.

**Table 1. Participant characteristics according to plain water intake.**

| | Total sample | Low intake | Moderate intake | High intake | P-value | $p_{trend}$ |
|---|---|---|---|---|---|---|
| | N = 25,369 | n = 8454 | n = 8455 | n = 8460 | | |
| Sex (% women) | 62.2 | 63.7 | 61.6 | 61.4 | 0.002 | 0.002 |
| Age (y) | 57 (51, 64) | 56 (51, 63) | 58 (51, 64)[a] | 57 (51, 64)[b] | < 0.001 | 0.012 |
| BMI (kg/m$^2$) | 25.1 (22.9, 27.7) | 25.0 (22.8, 27.5) | 25.1 (22.9, 27.7)[a] | 25.3 (23.0, 28.1)[a,b] | < 0.001 | < 0.001 |
| Apolipoprotein A1 (mg·L$^{-1}$) | 155 (138, 174) | 152 (136, 171) | 155 (139, 175)[a] | 157 (140, 177)[a,b] | 0.001 | < 0.001 |
| Apolipoprotein B (mg·L$^{-1}$) | 105 (88, 123) | 106 (89, 124) | 105 (89, 123)[a] | 103 (87, 121)[a,b] | < 0.001 | < 0.001 |
| Hypertension (%) | 60.0 | 58.5 | 60.6 | 60.9 | 0.002 | 0.001 |
| Currently smoke (%) | 28.4 | 32.3 | 26.2 | 26.9 | < 0.001 | < 0.001 |
| Lipid lowering medication use (%) | 2.1 | 1.8 | 2.2 | 2.3 | 0.044 | 0.021 |
| Dietary changers (%) | 21.3 | 15.5 | 20.8 | 27.7 | < 0.001 | < 0.001 |
| CAD incidence (%) | 14.7 | 15.8 | 14.4 | 13.9 | 0.002 | < 0.001 |
| Diabetes incidence (%) | 16.7 | 16.8 | 16.7 | 16.7 | 0.996 | 0.967 |
| Physical activity score (%): | | | | | | |
| Quintile 1 | 20.0 | 23.6 | 19.4 | 16.9 | < 0.001 | < 0.001 |
| Quintile 2 | 19.6 | 19.8 | 21.0 | 18.1 | | |
| Quintile 3 | 19.8 | 18.5 | 20.3 | 19.5 | | |
| Quintile 4 | 19.9 | 18.4 | 19.9 | 21.5 | | |
| Quintile 5 | 20.7 | 18.7 | 19.4 | 24.0 | | |
| Education level (%): | | | | | | |
| ≤ 8 y | 41.1 | 41.9 | 40.1 | 41.2 | < 0.001 | 0.159 |
| 9–10 y | 26.4 | 24.5 | 27.1 | 27.6 | | |
| 11–13 y | 17.9 | 19.1 | 17.6 | 17.1 | | |
| University degree | 14.6 | 14.6 | 15.2 | 14.1 | | |

Data are percentage in each category, or median (interquartile range). Data analysed using Chi square or Kruskal-Wallis with *post hoc* Bonferroni or Dunn Bonferroni adjustment, respectively

[a] *P*-value ≤ 0.001 compared to low

[b] *P*-value ≤ 0.033 compared to moderate

Table 2 shows a breakdown of dietary variables according to reported plain water intake. Non-water beverages were similar across tertiles of plain water intake, varying by ~100 g/d across quintiles, with similar variation in total water (sum of all water, including that obtained from food). Energy intake had a trend to decrease as plain water intake increased (2298 [IQR 1913, 2764] kcal/d at low plain water intake, decreasing to 2089 [IQR 1745, 2522] kcal/d at high plain water intake; $p_{trend} < 0.001$). Macronutrient composition was fairly stable across tertiles of plain water intake. Wholegrain, fish, fruit, and vegetable intake tended to increase with higher plain water intake (all $p_{trend} < 0.001$) (Table 2).

Fig 2 shows the correlation between different water intakes and diet variables. Plain and total water were strongly positively correlated ($r = 0.640$), as did non-water beverages and total water ($r = 0.718$). Healthy eating pattern (described previously [32]) was weakly positively associated with water intake ($r = 0.165$ plain water; $r = 0.175$ total water). Energy intake was weakly negatively correlated with plain water ($r = -0.130$) but weakly positively correlated with total water intake ($r = 0.322$). Fruit and vegetable intake were both weakly positively correlated to plain and total water intake, reflecting similar associations for fibre intake (Fig 2).

Table 3 shows that in unadjusted models, highest tertile of total water intake (HR 1.10, 95% CI 1.02, 1.19) but not highest tertile of plain water intake (HR 1.01, 95% CI 0.94, 1.09) was

**Table 2. Dietary variables according to plain water intake.**

| | Total sample | Low intake | Moderate intake | High intake | $P_{diff}$ | $p_{trend}$ |
|---|---|---|---|---|---|---|
| | N = 25,369 | n = 8454 | n = 8455 | n = 8460 | | |
| Plain water (g/d) | 564 (297, 945) | 199 (93, 298) | 564 (474, 662)[a] | 1140 (945, 1506)[a,b] | < 0.001 | - |
| Non-water beverages (g/d) | 1858 (1534, 2248) | 1913 (1586, 2323) | 1851 (1537, 2221)[a] | 1808 (1476, 2203)[a,b] | < 0.001 | < 0.001 |
| Total water (g/d) | 2517 (2086, 3041) | 2118 (1784, 2517) | 2421 (2103, 2806)[a] | 3071 (2645, 3572)[a,b] | < 0.001 | < 0.001 |
| EI (kcal/d) | 2194 (1829, 2640) | 2298 (1913, 2764) | 2194 (1851, 2619)[a] | 2089 (1745, 2522)[a,b] | < 0.001 | < 0.001 |
| EI reporter category (%): | | | | | | |
| Under | 15.1 | 13.3 | 13.9 | 18.2 | < 0.001 | < 0.001 |
| Adequate | 81.8 | 83.2 | 83.0 | 79.1 | | |
| Over | 3.1 | 3.5 | 3.1 | 2.7 | | |
| Healthy diet pattern (%)*: | | | | | | |
| Quintile 1 | 15.5 | 21.6 | 14.0 | 10.7 | < 0.001 | < 0.001 |
| Quintile 2 | 15.5 | 18.9 | 15.5 | 12.1 | | |
| Quintile 3 | 15.6 | 16.4 | 16.7 | 13.7 | | |
| Quintile 4 | 15.7 | 14.6 | 16.4 | 16.0 | | |
| Quintile 5 | 15.6 | 12.0 | 15.8 | 19.0 | | |
| Carbohydrate (%E) | 46 (42, 50) | 46 (42, 50) | 46 (42, 50)[a] | 47 (43, 51)[a,b] | < 0.001 | < 0.001 |
| Sucrose (%E) | 8 (6, 10) | 8 (6, 11) | 8 (6, 10)[a] | 8 (6, 10)[a,b] | < 0.001 | < 0.001 |
| Fibre (g/1000 kcal) | 9 (7, 11) | 8 (7, 10) | 9 (8, 11)[a] | 10 (8, 12)[a,b] | < 0.001 | < 0.001 |
| Fat (%E) | 38 (34, 42) | 39 (35, 43) | 39 (34, 42)[a] | 37 (33, 41)[a,b] | < 0.001 | < 0.001 |
| Saturated fat (%E) | 16 (14, 19) | 17 (14, 19) | 16 (14, 19)[a] | 16 (13, 18)[a,b] | < 0.001 | < 0.001 |
| MUFA (%E) | 13 (12, 15) | 14 (12, 15) | 13 (12, 15)[a] | 13 (11, 15)[a,b] | < 0.001 | < 0.001 |
| PUFA (%E) | 6 (5, 7) | 6 (5, 7) | 6 (5, 7)[a] | 6 (5, 7)[a,b] | 0.108 | < 0.001 |
| Protein (%E) | 15 (14, 17) | 15 (13, 16) | 15 (14, 17)[a] | 16 (14, 17)[a,b] | < 0.001 | < 0.001 |
| Wholegrain (portions/d) | 0.7 (0.3, 1.4) | 0.5 (0.2, 1.1) | 0.8 (0.3, 1.4)[a] | 0.9 (0.4, 1.6)[a,b] | < 0.001 | < 0.001 |
| Processed meat (g/d) | 32 (17, 52) | 36 (20, 57) | 32 (17, 52)[a] | 29 (15, 47)[a,b] | < 0.001 | < 0.001 |
| Fish (g/d) | 36 (20, 58) | 34 (17, 55) | 38 (21, 58)[a] | 38 (22, 60)[a] | < 0.001 | < 0.001 |
| Fruit (g/d) | 171 (106, 261) | 146 (86, 228) | 174 (111, 260)[a] | 200 (125, 291)[a,b] | < 0.001 | < 0.001 |
| Vegetables (g/d) | 163 (113, 227) | 146 (102, 204) | 163 (116, 226)[a] | 179 (124, 253)[a,b] | < 0.001 | < 0.001 |
| Alcohol (g/d) | 7 (2, 15) | 8 (2, 17) | 7 (2, 15)[a] | 6 (1, 13)[a,b] | < 0.001 | < 0.001 |
| Tea (g/d) | 32 (0, 225) | 21 (0, 225) | 43 (0, 225)[a] | 32 (0, 225)[b] | < 0.001 | 0.240 |
| Coffee (g/d) | 450 (250, 675) | 450 (300, 750) | 450 (250, 675)[a] | 450 (225, 675)[a] | < 0.001 | < 0.001 |
| SSB (g/d) | 8 (0, 94) | 29 (0, 129) | 21 (0, 94)[a] | 0 (0, 57)[a,b] | < 0.001 | < 0.001 |
| Fruit juice (g/d) | 1 (0, 100) | 0 (1, 100) | 0 (1, 114)[a] | 1 (0, 86)[b] | < 0.001 | 0.033 |
| Low fat milk (g/d) | 102 (0, 293) | 95 (0, 281) | 111 (0, 298)[a] | 100 (0, 304)[a] | < 0.001 | 0.003 |
| High fat milk (g/d) | 33 (11, 95) | 36 (13, 117) | 34 (12, 99)[a] | 30 (10, 75)[a,b] | < 0.001 | < 0.001 |

Data are percentage in each category, or median (interquartile range). Data analysed using Chi square or Kruskal-Wallis with *post hoc* Bonferroni or Dunn Bonferroni adjustment, respectively. Abbreviations: %E, percent total energy intake; EI, energy intake; MUFA, monounsaturated fatty acid; PUFA, polyunsaturated fatty acid; SSB, sugar-sweetened beverage. Plain water = tap + mineral water

Healthy diet pattern: n = 19,750 (low n = 7066; moderate n = 6629; high n = 6055)

[a] *P*-value < 0.05 compared to low

[b] *P*-value < 0.001 compared to moderate

associated with higher risk of type 2 diabetes compared to the lowest tertile of intake. After full adjustment, neither were associated with type 2 diabetes (both plain and total water HR 1.07, 95% CI 0.99, 1.16). No association between plain nor total water was found at moderate compared to low intake (Table 3 and S1 Fig).

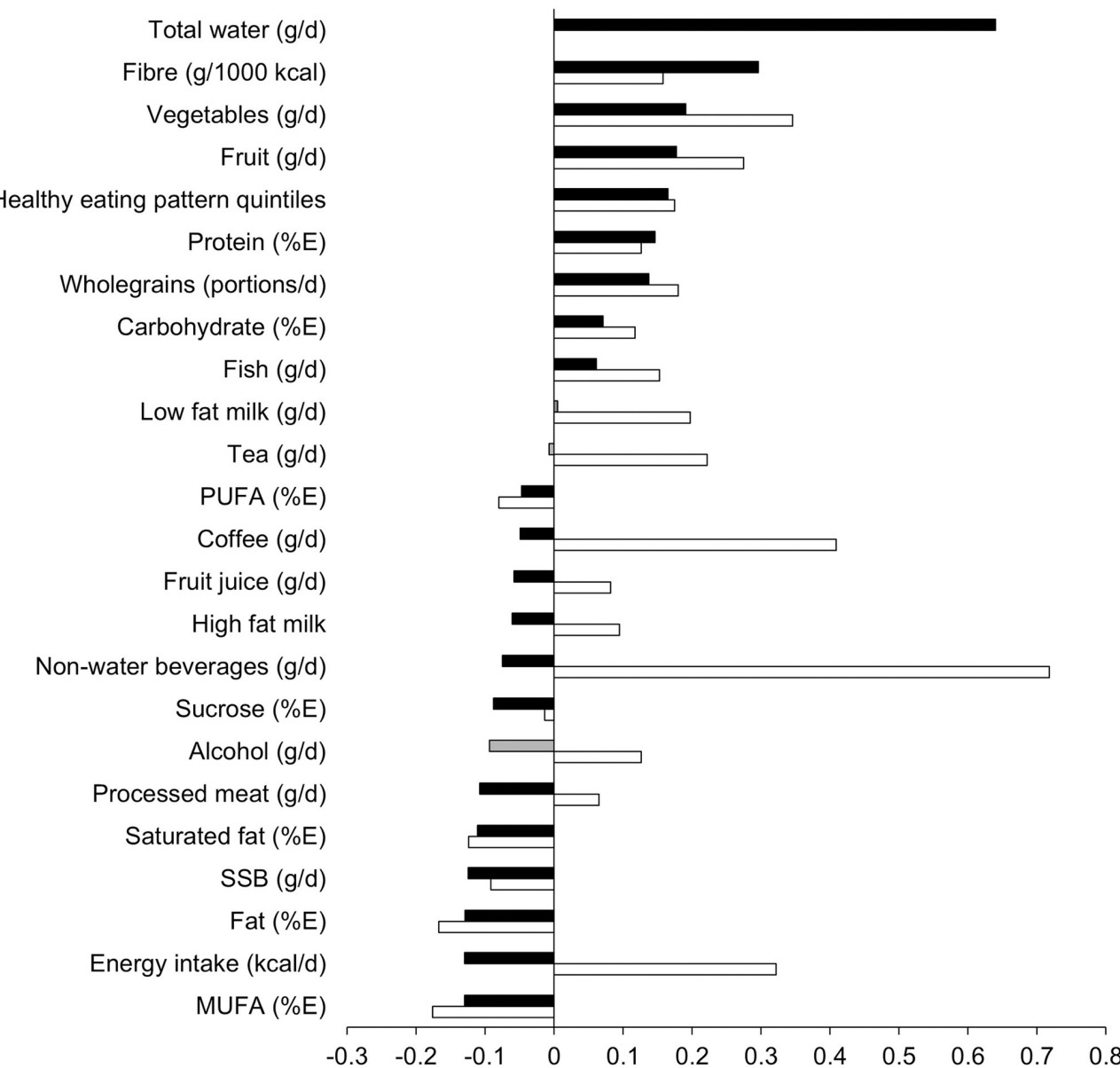

**Fig 2. Correlation coefficients between plain (black bars, grey if $p \geq 0.05$), and total (white bars) water intake and other dietary variables.** Data analysed using Pearson's. Abbreviations: %E, percent total energy; MUFA, monounsaturated fatty acid; PUFA, polyunsaturated fatty acids; SSB, sugar sweetened beverages.

In the unadjusted model, high intake of total water had no association with CAD (HR 0.99, 95% CI 0.92, 1.08) compared to low water intake (Table 3). Adjustment in model 2 demonstrated an increased associated risk (HR 1.13, 95% CI 1.04, 1.22), which remained similar through further adjustment, resulting in a 17% increased risk associated with consuming high *versus* low total water (most adjusted model HR 1.17, 95% CI 1.07, 1.27). In terms of plain water, no association was found between CAD and moderate (relative to low) intake (most adjusted model HR 1.02, 95% CI 0.94, 1.10). However, when comparing high to low plain water intake, there was a 12% reduction in CAD risk before adjustment (HR 0.88, 95% CI 0.82, 0.86), which reversed with adjustment (most adjusted model HR 1.13, 95% CI 1.04, 1.23)

**Table 3. Cox hazard model for tertiles of plain water and total water intake and type 2 diabetes and coronary artery disease.**

| | HR | 95% CI | *p* | HR | 95% CI | *p* |
|---|---|---|---|---|---|---|
| | **Plain water** | | | **Total water** | | |
| | Type 2 diabetes | | | | | |
| | Moderate (*versus* low) intake | | | Moderate (*versus* low) intake | | |
| Model 1 | 0.99 | 0.92, 1.06 | 0.748 | 1.03 | 0.96, 1.11 | 0.407 |
| Model 2 | 1.07 | 0.99, 1.15 | 0.090 | 1.03 | 0.96, 1.11 | 0.413 |
| Model 3 | 1.10 | 1.02, 1.18 | 0.018 | 1.08 | 1.00, 1.16 | 0.047 |
| Model 4 | 1.04 | 0.97, 1.12 | 0.291 | 1.03 | 0.96, 1.12 | 0.386 |
| Model 5 | 1.05 | 0.97, 1.13 | 0.202 | 1.04 | 0.96, 1.12 | 0.301 |
| | High (*versus* low) intake | | | High (*versus* low) intake | | |
| Model 1 | 1.01 | 0.94, 1.09 | 0.813 | 1.10 | 1.02, 1.19 | 0.011 |
| Model 2 | 1.15 | 1.07, 1.24 | < 0.001 | 1.12 | 1.04, 1.21 | 0.003 |
| Model 3 | 1.19 | 1.10, 1.28 | < 0.001 | 1.19 | 1.10, 1.28 | < 0.001 |
| Model 4 | 1.06 | 0.98, 1.14 | 0.167 | 1.05 | 0.97, 1.14 | 0.220 |
| Model 5 | 1.07 | 0.99, 1.16 | 0.075 | 1.07 | 0.99, 1.16 | 0.107 |
| | Coronary artery disease | | | | | |
| | Moderate (*versus* low) intake | | | Moderate (*versus* low) intake | | |
| Model 1 | 0.89 | 0.83, 0.97 | 0.005 | 0.96 | 0.89, 1.04 | 0.318 |
| Model 2 | 1.00 | 0.93, 1.08 | 0.993 | 1.00 | 0.92, 1.08 | 0.930 |
| Model 3 | 1.04 | 0.96, 1.12 | 0.357 | 1.03 | 0.95, 1.11 | 0.503 |
| Model 4 | 1.01 | 0.93, 1.09 | 0.834 | 1.03 | 0.95, 1.12 | 0.419 |
| Model 5 | 1.02 | 0.94, 1.10 | 0.662 | 1.04 | 0.96 1.13 | 0.335 |
| | High (*versus* low) intake | | | High (*versus* low) intake | | |
| Model 1 | 0.88 | 0.82, 0.96 | 0.002 | 0.99 | 0.92, 1.08 | 0.883 |
| Model 2 | 1.12 | 1.03, 1.21 | 0.008 | 1.13 | 1.04, 1.22 | 0.003 |
| Model 3 | 1.15 | 1.06, 1.25 | < 0.001 | 1.15 | 1.06, 1.25 | < 0.001 |
| Model 4 | 1.11 | 1.02, 1.20 | 0.013 | 1.15 | 1.06, 1.26 | 0.001 |
| Model 5 | 1.13 | 1.04, 1.23 | 0.004 | 1.17 | 1.07, 1.27 | < 0.001 |

Abbreviations: CI, confidence interval; HR, hazard ratio

Model 1: unadjusted

Model 2: model 1 + age, sex, diet method, season

Model 3: model 2 + smoking, alcohol intake, physical activity level, education

Model 4: model 3 + energy intake, energy intake misreporting, BMI, hypertension, lipid lowering medication, apolipoprotein A, apolipoprotein B

Model 5: model 4 + processed meat, wholegrains, and for plain water only: total water minus plain water

(Table 3 and S1 Fig). Sensitivity analyses did not yield majorly conflicting results; these are available in the Supplementary Material (S1 Table). Removing and/or stratifying by variables that violated proportional hazard assumptions yielded largely similar results, though some deviations were noted (S2–S5 Tables).

## Discussion

This analysis of > 25,000 Malmö residents showed both higher plain and total water intake to be associated with a marginally (12–17%) higher CAD risk compared to low water intake, after adjusting for relevant variables. These findings overall run contrary to previous studies which typically show plain water intake to either have no association with [9, 10], or lower risk of [8] cardiovascular outcomes. There was no association between plain nor total water intake and

type 2 diabetes risk after adjusting for relevant variates. These findings are in line with one study [6], but not others [3, 4, 11, 33]. Contrary to previous research [11], we did not find evidence of these results being moderated by sex. Our findings therefore did not support our hypothesis.

For type 2 diabetes, there seems to be a consistent change in results (for both plain and total water) when adding adjustment for energy intake, energy intake misreporting, BMI, hypertension, lipid lowering medication, apolipoprotein A, apolipoprotein B. This could suggest such factors are affecting the relationship; for example, higher BMI is a risk for type 2 diabetes, and by proxy of higher energy intake and fluid needs, more water is consumed. Considering the relationship between BMI and cardiovascular health, it is unclear why such a consistent trend was not found for CAD. However, our findings for type 2 diabetes are in accordance with an acute randomised controlled trial, finding hypohydration did not impact gluco-regulation [23], thus improving causal inference of our present results.

We ran extensive sensitivity analyses that nearly all agreed with our main analysis, improving the robustness of our findings (discussed further below). One potential explanation for our results is that higher fluid intakes can be caused by pathology, such as polydipsia in uncontrolled type 2 diabetes. If this were the case, it is unclear why this is not a consistent finding in the literature, and indeed we removed baseline type 2 diabetes prevalence to mitigate such an artefact. One reason is that such an association may in part be mitigated by those with healthier lifestyles also consuming higher water intake [34]. Indeed in our analysis, fruit, vegetables, and fibre were positively correlated with higher water intakes. Markers of healthy diet were adjusted for, suggesting an effect of water independent of these lifestyle markers. Including only those who were normotensive removed the association between CAD and plain water intake when comparing high *versus* low intakes.

Such findings may suggest those with primary pathology are driving the association despite controlling for this statistically—notwithstanding the reduction in sample size which may skew results. Accordingly, we excluded those in the top 10% of water intakes in sensitivity analyses, to determine the effects of pathological polydipsia. Most associations remained similar to the original findings. However, when comparing high *versus* low intakes, type 2 diabetes risk increased for both plain (HR 1.06, 95% CI 1.00, 1.18) and total (HR 1.11, 95% CI 1.00, 1.23) water intakes (relative to null findings in our original analysis). This suggests those who consumed the highest 10% of water intake were more likely doing so due to behavioural decisions rather than high intakes being pathologically driven; if the polydipsia was pathologically driven, excluding these participants would be expected to attenuate associations. Alternatively, salt sensitivity may play a role; i.e. high salt intake with high fluid retention leading to copeptin suppression [35]. We did not include salt in these analyses due to difficulties accurately determining intake based on the dietary intake methods. However, considering the correlation between water intake and healthy eating patterns (which are lower in salt), there is little evidence from these analyses support this hypothesis.

For plain water, both type 2 diabetes and CAD were associated with 2% higher risk in continuous models, which is discordant with both our original analyses (with null associations between moderate *versus* low intakes), though suggests the relationship may not be linear as the results are in line with the comparisons between high *versus* low intakes for CAD. Categorising continuous variables can lead to loss of power and less accurate estimations [36]. We categorised the variables to remove the influence of outliers on linear modelling, particularly in the lower and upper ends of intake, which are known to vary considerably [37]. Indeed, in our cohort, plain and total water intake ranged from 0 to 5.18 L/d and 0.8 to 9.0 L/d, respectively, and in the sensitivity analysis described above excluding the top 10% of consumers suggest the continuous model may not have been skewed by water intake variability.

Our findings largely go against current ideas regarding the cardiometabolic effects of hydration and health physiology. Whilst there is perhaps some debate regarding whether water intake/hydration status can directly improve health [13], there is no widely accepted hypothesis offering an explanation for chronic negative health effects. Some evidence suggests acute water intake prior to eating increases the postprandial glycaemic response, perhaps due to increased gastric emptying [38, 39]; however these effects are not consistent [40]. Additionally, if this was an underlying mechanism, we would expect to find an increase in type 2 diabetes risk primarily rather than an increased CAD risk.

One hypothesis relating water intake to cardiometabolic health is the lack of negative feedback loop when activating the hypothalamic-pituitary-adrenal axis via AVP [14]. We did not include AVP (or copeptin) in the present analyses as this was only measured in a subgroup of the cohort making comparisons difficult, and cortisol was not measured at all. However, this hypothesis posits that higher water intake would reduce AVP, thus mitigating the ill-effects of AVP-mediated cortisol elevation. Considering previous studies by our group, using the same cohort as the present analyses [18–21], show copeptin to be predictive of cardiometabolic ill-health, the current findings may suggest these associations may have been driven by pathologically and/or genetically higher copeptin. Accordingly, whilst water intake has been shown to reduce copeptin [22, 23], these reductions may only be as low as individual setpoints allow. Thus, although water intake can reduce copeptin (at least acutely), it may not be able to overcome other factors that more strongly determine (chronic) copeptin levels and health outcomes, offering a potential explanation as to why water intake was not associated with better health outcomes in the present study. This explanation does not offer insight into why water intake was associated with increased risk of CAD, however.

The measurement of fluid intake is a key limitation to these analyses. Fluid, and particularly plain water intake, is difficult to accurately recall and record [41]. Thus, measurement error may have confounded our analyses. Unlike energy intake, for which we can estimate and adjust for implausible reporting, to our knowledge, there is currently no equivalent method available for determine the predicted accuracy of fluid intake. In addition, whilst splitting water intakes into tertiles prevented issues such as sample size differences skewing the results, and whilst we believe we captured sufficient differences in plain water intake between low (199 mL/d) and high (1140 mL/d) intakes, it could be argued that the high intake group still were not consuming high enough intakes to induce meaningful physiological changes.

Whilst this study was a longitudinal study, our findings may still be indicative of reverse causation. Measures were taken at baseline, and such healthful behaviours may have been in response to diagnosis and/or greater awareness of ill-health. Excluding those with a type 2 diabetes or CAD diagnosis in the first two years after baseline measurements (in sensitivity analyses) did not meaningfully change the findings, reducing the likelihood (albeit not removing entirely) of reverse causality explaining our results. In addition, it is likely that drinking behaviours change over time, and we are unable to quantify this change in these analyses. Fluid intake typically decreases with ageing, particularly after 60 years [42]. Based on this, we can assume participants likely reduced their fluid intake during the study period, and assume this is roughly proportional between participants. However, further research should confirm or refute this, and our results should be taken within the context of this unknown.

Excluding those who changed their diet pattern did not change hazard estimates for low compared to moderate water intakes for type 2 diabetes, though the comparison between low and high plain water intakes resulted in an increased risk (high compared to low plain water: HR 1.10, 95% CI 1.01, 1.21; total water: HR 1.13, 95% CI 1.01, 1.26). Results for CAD and plain and total water remained similar (increased risk for low *versus* high intake only). Speculatively, this may suggest that post-baseline dietary changes were conducive to maintaining

gluco-regulatory health, and perhaps indicate some reverse causality such as those at higher risk improved their diet. It is unclear why these dietary changes would not similarly influence CAD risk though; such findings may be due to limitations such as loss of power in sensitivity analyses. Equally, we do not have data for post-baseline dietary changes.

We also found some assumption violations. Removing these variables from our model yielded some differences in results (S2 and S3 Tables), but it is unclear whether this was due to controlling for fewer variables or removing violations. To partially overcome this, we stratified our analyses by the offending variables, yielding largely similar results, though some differences were found (S4 and S5 Tables). Therefore some caution is required when interpreting our findings, and we emphasise the need for replication alongside mechanistic research.

In our analyses, plain and total water intake as the predictor variables yielded concordant results. Total water intake is more directly influenced by type and amount of food consumed, whereas plain water intake is primarily a behavioural decision. The agreement in our results between these two measures of water intake is interesting as they have many opposing correlations across other dietary variables, such as energy intake, processed meat, alcohol, and coffee. These dietary factors all may have the potential to independently influence CAD and type 2 diabetes. That our results were concordant between plain and total water perhaps suggests our statistical adjustment was sufficient to isolate effects of the water consumed.

Finally, our data are from a cohort of participants residing in Malmö, Sweden, between 1991–1995. The sampling method employed helped to improve representativeness but the generalisability of our results out with this cohort is unclear. Thus, our results may be an artefact of the study population, emphasising the need for similar work to replicate our analyses in other cohorts. That there were inconsistencies with some of our sensitivity analyses compared to our main analyses also raises the possibility that our findings are not reliable, further warranting more research.

Overall, we found that neither plain nor total water intake had an association with future type 2 diabetes risk, but when comparing high (not moderate) to low water intakes, CAD risk was increased by 12–17%. These findings are not explained by currently known mechanisms regarding hydration and health physiology, though should be taken in the context of the study design which cannot strongly infer causality. Future research should explore whether these findings are artefactual or have a mechanistic and causal explanation.

## Supporting information

**S1 Fig. Hazard ratios of the fully adjusted analyses across plain and total water and type 2 diabetes and coronary artery disease.** Data represent moderate or high tertiles of intake compared to low intake (reference). Error bars are 95% confidence intervals. * indicates HR is significant ($p \leq 0.05$). Black dots (●) indicate plain water and type 2 diabetes; empty dots (○) indicate total water and type 2 diabetes; black triangles (▲) indicate plain water and coronary artery disease; empty triangles (∆) indicate total water and coronary artery disease.
(TIF)

**S1 Table. Sensitivity analyses for type 2 diabetes and coronary artery disease.**
(DOCX)

**S2 Table. Type 2 diabetes analyses with variables that violate proportional hazard assumptions removed.**
(DOCX)

**S3 Table. Coronary artery disease analyses with variables that violate proportional hazard assumptions removed.**
(DOCX)

**S4 Table. Type 2 diabetes analyses stratified by variables that violate proportional hazard assumptions.**
(DOCX)

**S5 Table. Coronary artery disease analyses stratified by variables that violate proportional hazard assumptions.**
(DOCX)

## Author Contributions

**Conceptualization:** Olle Melander.

**Data curation:** Ulrika Ericson, Sofia Enhörning, Olle Melander.

**Formal analysis:** Harriet A. Carroll.

**Funding acquisition:** Harriet A. Carroll, Olle Melander.

**Investigation:** Harriet A. Carroll, Ulrika Ericson, Sofia Enhörning, Olle Melander.

**Methodology:** Harriet A. Carroll, Ulrika Ericson, Filip Ottosson, Olle Melander.

**Project administration:** Olle Melander.

**Supervision:** Olle Melander.

**Writing – original draft:** Harriet A. Carroll.

**Writing – review & editing:** Harriet A. Carroll, Ulrika Ericson, Filip Ottosson, Sofia Enhörning, Olle Melander.

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
