## [Decision Letter · Decision Letter 0]

24 Jul 2023

PONE-D-23-13468The association between water intake and future cardiometabolic disease outcomes in the Malmö Diet and Cancer Cardiovascular CohortPLOS ONE

Dear Dr. Carroll,

Thank you for submitting your manuscript to PLOS ONE. After careful consideration, we feel that it has merit but does not fully meet PLOS ONE’s publication criteria as it currently stands. Therefore, we invite you to submit a revised version of the manuscript that addresses the points raised during the review process.

Please address the concerns brought up by the reviewers.

We look forward to receiving your revised manuscript.

Kind regards,

William M. Adams

Academic Editor

PLOS ONE

Journal Requirements:

“This project was funded by the Esther Olssons stiftelse II & Anna 429 Jönssons Minnesfond (HC). The funders had no role in study design, data collection 430 and analysis, decision to publish, or preparation of the manuscript.”

“This project was funded by the Esther Olssons stiftelse II & Anna Jönssons Minnesfond (HC). The funders had no role in study design, data collection and analysis, decision to publish, or preparation of the manuscript.”

“HAC has received research funding from the Economic and Social Research Council, the European Hydration Institute, and the Esther Olssons stiftelse II & Anna Jönssons Minnesfond; has conducted research for Tate & Lyle; and has received speakers fees from Danone Nutricia Research. HAC, SE and OM have received conference, travel and accommodation fees from Danone Nutricia Research. O.M. has received a research grant for another study and consultancy fees from Danone Nutricia Research. No other authors declare any conflicts of interest.”

Reviewers' comments:

Reviewer's Responses to Questions

**Comments to the Author**

1. Is the manuscript technically sound, and do the data support the conclusions?

Reviewer #1: Yes

Reviewer #2: Partly

2. Has the statistical analysis been performed appropriately and rigorously? 

Reviewer #1: Yes

Reviewer #2: I Don't Know

3. Have the authors made all data underlying the findings in their manuscript fully available?

Reviewer #1: Yes

Reviewer #2: Yes

4. Is the manuscript presented in an intelligible fashion and written in standard English?

Reviewer #1: Yes

Reviewer #2: No

5. Review Comments to the Author

Reviewer #1: This study provided a longitudinal analysis on the association between baseline water intake and incidence of CAD and type 2 diabetes among a large cohort using cox proportional hazards models. This study adds to the existing literature on hydration and health, with results suggesting an association between water and CAD risk (opposite to the expected relationship) but not type 2 diabetes risk in this cohort. I have a few suggestions which may enhance the quality of this paper.

• The cox proportional hazards analyses include quite a lot of covariates for this type of model despite the large sample size. Can you please provide the concordance statistics as some indication of model fit changes with the inclusion or omission of some of these variables? Additionally, given the complexity with these additional variables, can you confirm if the models met the proportional hazards assumptions for each covariate?

• Can you provide more rationale for the covariates used in the models?

• Line 175 and Line 260 – change “waiter” to “water”

• Can you comment somewhere on the expected persistence (or changes) in the fluid intake and dietary behaviors over the course of the ~20 years until follow up?

• Thank you for including a continuous model as well!

Reviewer #2: 1) The manuscript is not well written and in many sections is ambiguous.

2) Cardiometabolic factors are mentioned in the title of the article, but only diabetes and CAD have been investigated.

3) The final results of the study do not seem logical. According to the amount of water intake reported in the participants in table 2, the mechanism proposed in the discussion section is not correct (page 18, line 303). According to Table 1, the incidence of CAD was lower in subjects who received more water, while the final result of this study shows the opposite.

4) While appreciating the efforts of the authors, it seems that measurement error, out of date data, and the way of statistical analysis have caused disruption in the results.

Sincerely yours

6. PLOS authors have the option to publish the peer review history of their article (what does this mean?). If published, this will include your full peer review and any attached files.

Reviewer #1: No

Reviewer #2: No

---

## [Author Response · Author response to Decision Letter 0]

14 Oct 2023

Thank you to the reviewers for taking the time to assess our manuscript. Our reply contains tables so we have attached a Word document with our full response. We hope we have adequately addressed all points raised.

---

## [Decision Letter · Decision Letter 1]

22 Nov 2023

PONE-D-23-13468R1The association between water intake and future cardiometabolic disease outcomes in the Malmö Diet and Cancer Cardiovascular CohortPLOS ONE

Dear Dr. Carroll,

Thank you for submitting your manuscript to PLOS ONE. After careful consideration, we feel that it has merit but does not fully meet PLOS ONE’s publication criteria as it currently stands. Therefore, we invite you to submit a revised version of the manuscript that addresses the points raised during the review process.

We look forward to receiving your revised manuscript.

Kind regards,

William M. Adams

Academic Editor

PLOS ONE

**Additional Editor Comments:**

Please address the comments below from the reviewers.

Reviewers' comments:

Reviewer's Responses to Questions

**Comments to the Author**

1. If the authors have adequately addressed your comments raised in a previous round of review and you feel that this manuscript is now acceptable for publication, you may indicate that here to bypass the “Comments to the Author” section, enter your conflict of interest statement in the “Confidential to Editor” section, and submit your "Accept" recommendation.

Reviewer #1: All comments have been addressed

Reviewer #3: (No Response)

Reviewer #4: (No Response)

2. Is the manuscript technically sound, and do the data support the conclusions?

Reviewer #1: Yes

Reviewer #3: Partly

Reviewer #4: Partly

3. Has the statistical analysis been performed appropriately and rigorously? 

Reviewer #1: Yes

Reviewer #3: Yes

Reviewer #4: I Don't Know

4. Have the authors made all data underlying the findings in their manuscript fully available?

Reviewer #1: Yes

Reviewer #3: Yes

Reviewer #4: Yes

5. Is the manuscript presented in an intelligible fashion and written in standard English?

Reviewer #1: Yes

Reviewer #3: Yes

Reviewer #4: Yes

6. Review Comments to the Author

Reviewer #1: Thank you for addressing my comments. Please note very minor grammatical error in line 60, I believe this should be "mediated". Otherwise, I appreciate you incorporating my suggestions into the manuscript and am happy to accept this manuscript.

Reviewer #3: Thank you for your great study and I am sure you had to put a lot of efforts.

In general, methods are too vague. More details are required.

Line 103-106: Precise information is required here, including how to process this data, how to assess plain and total water intake by these measurements, how to capture water intake outside of meals, how 1,2, and 3 methods used to calculate total water intake etc.

Line 146: What is the justification to exclude participants who had a history of CAD or type 2 diabetes from analyses? This might change the results as the number of CAD and diabetes is intentionally decreased, and water intake may already (or may not) associate with them at baseline for some participants.

Line 154: what is the cut point of low, moderate, and high intake, and the citation is required here.

Line 182-188: Justifications and citations are required for each cut point for sensitivity.

Line 190: When is referring to follow-up time?

Line 194-196: Even 1140 g/d can be categorized as “low drinker” in some previous studies, so these cut points might not capture results appropriately. At least, it is not appropriate to call “high” at this level.

Table 1 &2: Need to reorganize as it is hard to see.

Discussion needs to be changed accordingly.

Reviewer #4: Thank you for the opportunity to review the current manuscript regarding the relationship between recorded water intake and longitudinal risk for development of T2DM and CAD within a population of the Malmo Diet and Cancer Cohort. There are a few questions regarding the statistical approach that I hope the authors can provide clarity for.

It appears that the collinearity between the data used to group participants into tertiles (i.e., plain water or total water) and the covariate of model 5 (plain water) would limit the ability of the grouping to account for any other variation in the dependent variables (i.e., T2DM or CAD). Additionally, the description of the "plain water only" variable as total water minus plain water is confusing because by that math, the difference would be every source of water intake except plain water. If the covariate was indeed "other water sources" that could answer the question if T2DM or CAD risk differs by plain water tertile, however it does not address the underlying physiological question/hypothesis related to water intake and disease.

The authors mention that the purpose of converting a continuous variable into a categorical variable was done so to reduce the impacts of outliers. However, the concerns of such practice remain as outlined in the following reference, Howard Wainer, Marc Gessaroli & Monica Verdi (2006) Visual Revelations, CHANCE, 19:1, 49-52, DOI: 10.1080/09332480.2006.10722771

There is also mention of a continuous model on line 338 of the Discussion, however I am not able to locate this analysis within the methods or results section.

The results of this investigation are compelling as they do not agree with previous research in the area. The authors have devoted considerable space within the current manuscript to controls and checks which should enhance the readers confidence in the outcomes. Addressing the two concerns above will provide further confidence for the reader when reading something they may not have previously believed to be the case and avoid the misappropriated sentiment of "Well, this is just a case of fancy p hacking."

7. PLOS authors have the option to publish the peer review history of their article (what does this mean?). If published, this will include your full peer review and any attached files.

Reviewer #1: No

Reviewer #3: No

Reviewer #4: No

---

## [Author Response · Author response to Decision Letter 1]

7 Dec 2023

We have submitted a Word document with our full response to reviewers.

---

## [Decision Letter · Decision Letter 2]

19 Dec 2023

The association between water intake and future cardiometabolic disease outcomes in the Malmö Diet and Cancer Cardiovascular Cohort

PONE-D-23-13468R2

Dear Dr. Carroll,

We’re pleased to inform you that your manuscript has been judged scientifically suitable for publication and will be formally accepted for publication once it meets all outstanding technical requirements.

Kind regards,

William M. Adams

Academic Editor

PLOS ONE

Additional Editor Comments (optional):

Reviewers' comments:

Reviewer's Responses to Questions

**Comments to the Author**

1. If the authors have adequately addressed your comments raised in a previous round of review and you feel that this manuscript is now acceptable for publication, you may indicate that here to bypass the “Comments to the Author” section, enter your conflict of interest statement in the “Confidential to Editor” section, and submit your "Accept" recommendation.

Reviewer #3: All comments have been addressed

Reviewer #4: (No Response)

2. Is the manuscript technically sound, and do the data support the conclusions?

Reviewer #3: Yes

Reviewer #4: Yes

3. Has the statistical analysis been performed appropriately and rigorously? 

Reviewer #3: Yes

Reviewer #4: I Don't Know

4. Have the authors made all data underlying the findings in their manuscript fully available?

Reviewer #3: Yes

Reviewer #4: Yes

5. Is the manuscript presented in an intelligible fashion and written in standard English?

Reviewer #3: Yes

Reviewer #4: Yes

6. Review Comments to the Author

Reviewer #3: No more comments. Thank you for addressing previous comments. I believe this paper will add information to the field.

Reviewer #4: Thank you for the edits and clear explanations. My one remaining point of confusion is the terminology within Table 3 and the associated text. Specific to Model 5, the explanation provided states that this model controlled for "total water minus plain water", however the label given to this variable is "plain water only", while the actual variable is all sources of water except plain water. There must be a rationale for the "plain water only" label that I am not understanding. I assume that if I'm having a hard time, the readers may as well.

7. PLOS authors have the option to publish the peer review history of their article (what does this mean?). If published, this will include your full peer review and any attached files.

Reviewer #3: No

Reviewer #4: No

---

## [Editor Report · Acceptance letter]

10 Jan 2024

PONE-D-23-13468R2 

PLOS ONE

Dear Dr. Carroll, 

I'm pleased to inform you that your manuscript has been deemed suitable for publication in PLOS ONE. Congratulations! Your manuscript is now being handed over to our production team.

Kind regards, 

on behalf of

Dr. William M. Adams 

Academic Editor

PLOS ONE